# The Patients’ Long-Term Insight into Day-to-Day Functioning After Establishing the Functional Seizures Diagnosis

**DOI:** 10.3390/brainsci15020120

**Published:** 2025-01-26

**Authors:** Jelena Lazovic, Ognjen Radojicic, Ivo Bozovic, Aleksa Pejovic, Dragoslav Sokic

**Affiliations:** 1Neurology Clinic, University Clinical Centre of Serbia, 11000 Belgrade, Serbia; jelena.lazovic5@gmail.com (J.L.); ivo.bozovic20@gmail.com (I.B.); aleksa_dartmoor@live.com (A.P.); 2Faculty of Medicine, University of Belgrade, 11000 Belgrade, Serbia; ogi.radojicic@gmail.com

**Keywords:** functional seizures, QoL, everyday functioning, depressive symptoms, FS perception

## Abstract

Background/Objectives: Functional seizures (FSs) are paroxysmal, time-limited events with motor, sensory, autonomic, or cognitive manifestations related to pathophysiological processes other than abnormal electric discharges in the brain. However, these seizures are often followed by different psychiatric comorbidities. Their impact on the overall quality of life and the cofounding factors, especially the ones that can be treated, were the main investigation aims of this study. Methods: This study comprised 76 patients who were diagnosed with FSs. This study included patients who were diagnosed with FSs via video-EEG telemetry. We used the “Likert scale” from the QOLIE 31 questionnaire to evaluate patients’ subjective perception of their quality of life. We researched the association of various clinical factors with the subjective QoL score. Results: A statistically significant marginal association was shown for seven variables, four of them with a positive association (subjective perception of disease severity, belief in treatments’ positive effects, functional seizure cessation, and not being afraid of next seizure) and three of them with a negative association (age at FS onset, depression symptoms, and current age of life). After Bonferroni correction for multiple testing only symptoms of depression stayed statistically significantly associated with outcome. Multivariable logistic regression following variable selection identified that six variables (age at FS onset, absence of psychiatric testing, perceiving oneself as mentally changed due to the disease, seizure severity, depressive symptoms, and fear of therapy side effects) were statistically significantly negatively associated with the outcome. Conclusions: It seems that patients who have FSs coexisting with depressive symptoms and also those with worse disease perception have less chances to be satisfied with their overall quality of life.

## 1. Introduction

One of the most common reasons for functional disability among patients suffering from different neurologic diseases are functional neurological disorders [1]. Functional seizures (FSs) might be the most frequent subtype of functional neurological disorders, superficially resembling epileptic seizures due to their sudden and involuntary appearance [1,2]. Typical FS presentation is comprised of a paroxysmal, time-limited event with different motor, behavioral, and cognitive manifestations [1,2,3]. Although the precise incidence of FSs still remains undetermined, the literature to date underlines that up to 40% of all patients admitted for continuous video-EEG monitoring are finally diagnosed with FSs [4]. Moreover, recent data have shown that about 10% of patients presenting as benzodiazepine-refractory generalized convulsive status epilepticus was found to have a FS episode. Patients suffering from FSs are usually women in their twenties or thirties, having a cooccurring psychiatric disorder in 70% of cases. More than 30% of psychiatric disorders are from the dissociative spectrum [5,6].

Although the exact pathophysiological mechanism of FSs remains controversial, several theories have been reported. The main focus of these studies is on the disequilibrium between the emotional, somatic, and motor systems [4]. Among them, the integrative cognitive model stands up as the most prominent one, but further evaluation is still needed in order to prove these suspicions.

Health-related quality of life (QoL) is a multidimensional concept encompassing different aspects of physical, mental, and social well-being [7]. Psychogenic nonepileptic events might affect one’s overall QoL. The chronic course of the disease consequently causes significant economic burden for society [8]. Moreover, the quality of life is often lower in patients with FSs compared to epilepsy patients [9].

The main aim of our study is the analysis of self-reported QoL in patients at least one year after establishing a diagnosis of psychogenic nonepileptic seizures.

## 2. Material and Methods

### 2.1. Patients

This study initially comprised 174 patients, identified from our medical database, who were further contacted by telephone starting from July 2022 and at least one year after the diagnosis had been established (preferably more than a year in order to obtain relevant long-term information). Only FSs which were diagnosed using video-EEG telemetry were included in this research. The data were collected from the local hospital’s informative systems “Infomedis” and “Heliant” and all patients with FSs from 2009 to 2020 were included. The data were obtained from the patients and their caregivers.

From the initial number of 174 identified patients, 3 patients were excluded due to the presence of malignancy, while 1 patient was excluded because of the presence of chronic alcoholism. Four patients were not included because they died (not related to functional seizures). Unfortunately, we were not able to make contact with 90 patients. Thus, this study comprised a final number of 76 patients with definite FSs, diagnosed using video-EEG telemetry, which includes 96 h of video EEG monitoring, magnetic resonance brain imaging and neuropsychological testing (Figure 1). This study was approved by the Ethical Board of the Neurology Clinic, University Clinical Center of Serbia. All patients or their caregivers gave informed consent to participate in the study.

We used the “Likert scale” from the QOLIE 31 questionnaire to evaluate patients’ subjective perception of their quality of life. The “Likert scale” ranges from 0 to 10 (0 being the worst possible QoL, while 10 being the best). Our patients or their caregivers rated the patient’s quality of life by circling one number on the scale, where a score of ≥7 was considered as a good QoL [7].

Different sociodemographic and diagnostic data were also collected from patients’ medical records, patients themselves, and caregivers at time of analyzing. The obtained information included personal information, current disease status and treatment, and significant comorbid disorders.

### 2.2. Statistical Analysis

All data processing, modeling, and analyses were performed using R (version 4.1.3) [10]. For comparisons between groups, the Student’s t-test was used for normally distributed continuous variables, the Mann–Whitney U test was applied to non-normally distributed continuous variables, and the Chi-square test was utilized for categorical variables.

Univariate logistic regression was performed to analyze marginal association of each variable with good QoL. Bonferroni correction was applied to adjust for multiple testing. Cross-validated Lasso (Least Absolute Shrinkage and Selection Operator) logistic regression was employed for variable selection using the glmnet package in R, with the optimal lambda (the minimal lambda) determined via 10-fold cross-validation [11]. This approach reduced dimensionality by identifying a subset of variables collectively associated with the outcome. Finally, multivariable logistic regression with the selected variables was used to identify factors conditionally associated with a QoL score greater than 7. All hypothesis tests were two-tailed, with the significance threshold set at a *p*-value of 0.05. All regression analyses applied in our study are shown in Appendix A.

## 3. Results

Our study finally comprised 76 analyzed patients diagnosed with FSs. There was no statistically significant difference in gender distribution, current patient’s age, and the age at diagnosis of FSs between the analyzed cohort and the group of patients without an established telephone contact.

In our study population, 62 of the patients (84%) were females (male to female ratio 1:4.5, respectively). The mean onset of PNES was 25.08 years (SD: 14.05), the mean age at PNES diagnoses was 34.37 years (SD: 13.61), and the mean diagnostic delay was 9.41 years (SD: 8.8). All variables had similar distributions by gender (Table 1).

We first performed univariate logistic regression to see the independent association of all clinical and demographical variables with good QoL. A statistically significant marginal association was shown for 7 variables, 4 of them with a positive association (subjective perception of disease severity, belief in treatments’ positive effects, functional seizure cessation, and not being afraid of next seizure) and 3 of them with a negative association (age at FS onset, depression symptoms, and current age of life). After Bonferroni correction for multiple testing, only depression symptoms remained statistically significantly associated with outcome (Table 2).

For variable selection, Lasso logistic regression was performed, identifying 16 out of 43 clinical and demographic variables as jointly associated with good QoL. A multivariable logistic regression model was then refitted using the selected 16 variables, revealing that 6 variables (age at FS onset, absence of psychiatric testing, perceiving oneself as mentally changed due to the disease, seizure severity, depression symptoms, and fear of therapy side-effects) were statistically significantly associated with the outcome (Table 3). Notably, all six variables were negatively associated with the outcome, as indicated by odds ratios less than 1.

## 4. Discussion

The aim of our explorative study was to emphasize subjective measures of the day-to-day functioning and the overall QoL value at least one year after diagnosis and investigate the f factors that influence the subjective values of the QoL and clinical factors, regarding the course and the clinical characteristics of the disease.

In our study, the population was mostly female and in their third decade, which is consistent with the data that FSs are more frequent in females, with a 3:1 ratio. Although this inconsistent difference cannot be fully understood, the gender-related different brain region connectivity, especially including the central region for emotional processing—the left amygdala and the dorso-medial prefrontal cortex—could serve as an explanation for this female predominance. The factors also contributing to the altered negative emotional processing are biological (sex hormones) and psychosocial (gender role and identity). Further, any negative experience from early childhood seems to be more prevalent in woman and alters the brain connectivity influencing the emotional processing, with this considered the biological basis for the development of psychogenic nonepileptic seizures [12]. Moreover, the mean age of onset in our study was around 25, which is in accordance with the mean age of onset of 27.5 years in the literature data to date [13].

According to the literature, quality of life can be exerted on by social characteristics, comorbidities, treatment modalities, and the frequency of psychogenic nonepileptic spells and their eventual complete cessation, also bearing in mind the social circumstances of the patient, the perception of the disease itself, and the patient’s belief that the treatment could help them [14].

### 4.1. Personal Factors and the Belief About the Disease

Older patients have fewer chances to be satisfied with their day-to-day functioning, as expressed in QoL values above 7. In the literature to date, older age is one of the main predictors of decreased QoL, which is probably due to the effect of chronic conditions in older subjects. On the other hand, depression is also typically more frequent in elderly compared to younger individuals. The negative impact of depression and older age has been proved also in other neurological disorders [15].

According to the results, if the patient thought that the disease was very severe, the possibility of reaching a satisfactory QoL value was lower than for those who considered their illness to be not as serious.

In patients whose belief in treatment was adequate and who had confidence that treatment could help in overcoming the disease’s impediments, their QoL was better. Further, the people who feared therapy side effects and questioned their efficacy had less chance to reach a satisfactory score of QoL (more than 7). The patient’s perception about the disease itself and also treatment strategies has been shown to have an impact on the possibility of reaching a satisfactory QoL, which is consistent with the literature data that describe the influence of disease perception on the conversion diseases and its dynamics [16].

As one of the perpetuating factors, depression and its symptoms (depressive mood, crying often) have been shown to be one of the most important contributors to the low quality of life of people suffering from psychogenic nonepileptic seizures [7,8,9]. Therefore, depression and its symptoms have to be prioritized since the amelioration of these symptoms improves their QoL significantly [17,18,19]. It is important to treat depressive symptoms with proper therapy as soon as the symptoms are recognized. The significance of this conclusion is even clearer when bearing in mind that the treatment of this comorbidity is the only formalized treatment in curing psychogenic nonepileptic spells [13].

### 4.2. Clinical Factors and the Characteristics of the Disease

Although it could be suggested that a longer disease duration might have a negative impact on QoL due to the burden of a chronic disease, our results indicated that later disease onset is correlated with worse subjective QoL perception. The explanation could lie in the fact that the younger population has better coping strategies, as compared to older people having less plasticity in improving their functional status in everyday life. These data are congruent with the current literature data [20]. Our findings indicate that the first years after being diagnosed with FSs could be the most demanding for patients, and clinicians should be aware of this.

In patients with persisting FSs, the frequency of attacks was not associated with QoL and their reduction did not have predictive value [9], although psychogenic nonepileptic seizures’ complete cessation, according to our results, has an impact on QoL. Patients who do not experience seizures anymore have a greater probability of reaching a subjective QoL value above 7. Absolute FS cessation, or being spell-free, makes the life quality better, justifying the therapeutic effort to achieve complete freedom from psychogenic nonepileptic seizures [14].

Pharmacological treatment in psychogenic epileptic seizures has no benefit, so the withdrawal of antiepileptic medication is recommended, especially bearing in mind the potential side effects and unnecessary costs. Some of our patients took antiseizure medications and the ASM did not show any clear consequences of drug toxicity and had no impact on the quality of life, which is not concordant with the literature data [17]. The explanation probably lies in the fact that our patients did not take the pharmacological treatment because it was not prescribed or were stopped early in the disease course since there are no evidence-based treatments for the cessation of psychogenic nonepileptic seizures.

## 5. Conclusions

It seems that patients who have FSs coexisting with depressive symptoms and also those with worse disease perception have a lower likelihood of being satisfied with their overall quality of life.

## 6. Limitations

The main limitation of our study is its small sample size, since we were not able to make telephone contact with 90 patients. Further, we have not used a disease-specific QoL questionnaire, which would be the next step in our research.

## Figures and Tables

**Figure 1 brainsci-15-00120-f001:**
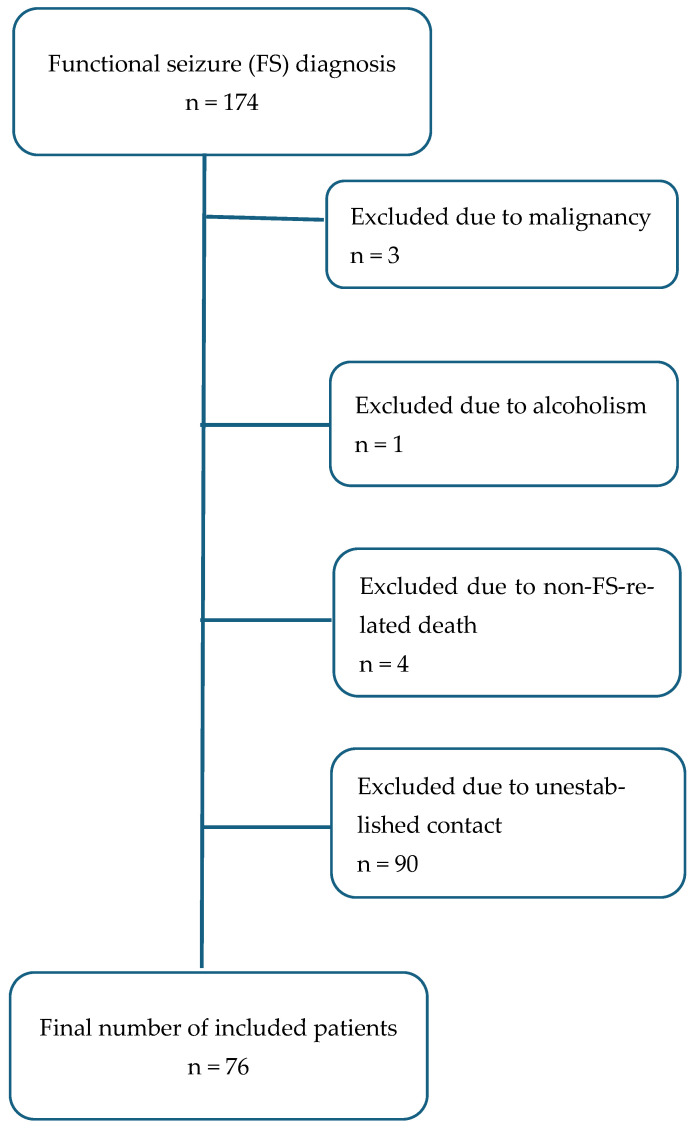
Flowchart representing included and excluded FS patients.

**Table 1 brainsci-15-00120-t001:** Demographic and clinical characteristics of FS patients (N = 76).

Gender	Females	Males	*p*
N (%)	62 (83.8%)	14 (16.2%)	
Age (mean ± SD, years)	42.5 ± 12.79	39 ± 13.47	0.363
Age of first attack (mean ± SD, years)	26.08 ± 13.61	18.5 ± 13.01	0.062
Time of diagnosis (mean ± SD, years)	8.42 ± 8.01	12.57 ± 10.23	0.102
Functional seizure cessation (n, %)	14 (23.7%)	3 (21.4%)	1
Having comorbidity	47 (75.8%)	13 (92.9%)	0.293
Education None/primary/high school (%)	50 (80.6%)	12 (85.7%)	
More than high school (%)	12 (19.4%)	2 (14.3%)	

SD—standard deviation.

**Table 2 brainsci-15-00120-t002:** Association of clinical and demographic variables with subjective QoL values.

Variable	Odds Ratio of Having QoL Greater Than 7 (95% CI)	*p* Value
Age at FS onset	0.957 [0.917–0.995]	0.0283 *
Subjective perception of disease severity	4.55 [1.65–12.5]	0.00332 *
Belief in treatment’s positive effects	3.36 [1.24–9.06]	0.0168 *
Functional seizure cessation	5.38 [1.75–18.10]	0.00429 *
Depression symptoms	0.102 [0.030–0.343]	0.000226 **
No fear of next seizure	3.37 [1.23–9.23]	0.0179 *
Current age of life	0.938 [0.897–0.980]	0.00448 *

The results are presented as odds ratio and 95% confidence interval. The criteria for the significance of statistical differences were * *p* < 0.05, ** *p* < 0.05 after Bonferroni correction.

**Table 3 brainsci-15-00120-t003:** Association of clinical and demographic variables with QoL measures.

	Odds Ratio (95% CI)	*p* Value
Neurological comorbidity	1.0245 (0.0399–26.35)	0.9881
Age at FS onset	0.788 (0.642–0.9665)	0.0221 *
Perceiving oneself as mentally changed	0.00086 (0.00–0.9602)	0.0486 *
Changing frequency	0.00 (0.00–100)	0.9983
Absence of psychiatric testing	0.0003 (0.00–0.254)	0.018 *
FS cessation	3.54 (0.246–50.93)	0.3522
Severity of seizure	0.001 (0.00–0.4711)	0.0277 *
Dystonia symptom as manifestation	0.00 (0.00–100)	0.9968
Sensory symptoms as manifestation	59.84 (0.08–44128)	0.2245
Taking the treatment regularly—no	14.83 (0.536–410.38)	0.111
Hospital staying	0.00 (0.00–100)	0.9945
No fear of next seizure	19.55 (0.7204–530.79)	0,0775
Fear of therapy side-effects	0.0007 (0.00–0.646)	0.037 *
Depression symptoms	0.004 (0.000–0.309)	0.0127 *
Lower number of attacks	2.69 (0.1606–45.075)	0.4912
Age of life	1.031 (0.875–1.21)	0.7132

* *p* < 0.05.

## Data Availability

The data that support the findings of this study are available on request from the corresponding author. The data are not publicly available due to privacy or ethical restrictions.

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
