# Peer review of "The Patients’ Long-Term Insight into Day-to-Day Functioning After Establishing the Functional Seizures Diagnosis"

_brainsci, 2025, doi:10.3390/brainsci15020120_

Round 1
Reviewer 1 Report
Comments and Suggestions for Authors
In this paper, the authors investigated the association between various clinical and lifestyle factors and quality of life. The study included 76 patients with psychogenic non-epileptic seizures (PNES) diagnosed through video-EEG telemetry. The findings revealed that patients with PNES who also exhibit depressive symptoms, as well as those with a poorer perception of their condition, are less likely to report satisfaction with their overall quality of life. The study is well-conducted, and I have no comments, except for reference no. 11, which lists only 12 out of the 38 authors.
Author Response
Comments 1: In this paper, the authors investigated the association between various clinical and lifestyle factors and quality of life. The study included 76 patients with psychogenic non-epileptic seizures (PNES) diagnosed through video-EEG telemetry. The findings revealed that patients with PNES who also exhibit depressive symptoms, as well as those with a poorer perception of their condition, are less likely to report satisfaction with their overall quality of life. The study is well-conducted, and I have no comments, except for reference no. 11, which lists only 12 out of the 38 authors.
Response 1: Thank you for your kind comments and suggestions. We have made the suggested correction in the "Reference" section, highlighted in red.
Reviewer 2 Report
Comments and Suggestions for Authors
The main aim of the study is reported as the analysis of self-reported QoL in patients at least one year after establishing the diagnosis of psychogenic non epileptic seizures.
These self-reported QoL data including personal information, current disease status and treatment, significant comorbid disorders and QoL interpretation were obtained from the patients and their caregivers via call phone. Other different sociodemographic and diagnostic data were collected from patients’ medical records, patients themselves and caregivers at time of analyzing.
In our opinion, there are some outstanding methodological issues including:
- Unclear self-reported QoL variables in terms of their definition and extraction methods from patients and caregivers interview;
- Some materials used are unclear, e.g.: “The use of the Likert scale from the QOLIE-31 questionnaire about the patient’s subjective perception of their quality of life” is mentioned in the abstract however in the section “Materials and methods” is unclear how this Likert scale was administered to the patients and the variables referred to;
- Some statistical analysis procedures are unclear, e.g.:
1. We read that the association of various clinical and lifestyle factors with the subjective QoL score were researched. What QoL subjective score was used for the regression analysis and why a score greater>7 was considered as a satisfactory QoL score?
Finally, the PNES term has currently been replaced in the literature with the “Functional seizures (FS)” term, we suggest to use this last proposed terminology in your paper.
Author Response
Comments 1: “The main aim of the study is reported as the analysis of self-reported QoL in patients at least one year after establishing the diagnosis of psychogenic non epileptic seizures.
These self-reported QoL data including personal information, current disease status and treatment, significant comorbid disorders and QoL interpretation were obtained from the patients and their caregivers via call phone. Other different sociodemographic and diagnostic data were collected from patients’ medical records, patients themselves and caregivers at time of analyzing.
In our opinion, there are some outstanding methodological issues including:
- Unclear self-reported QoL variables in terms of their definition and extraction methods from patients and caregivers’ interview”
Response 1: Thank you for this useful question.
The self – reported values of the QoL were extracted from the QOLIE 31 questionnaire - its part named the “Likert scale”. The “Likert scale” ranges from 10 to 0 (10 being the best possible QoL and 0 being the worst). Our patients or caregivers rated the patient’s quality of life by circling one number on the scale.
We have improved the “Abstract” and the “Material and methods” part of the manuscript with more clear information, which are highlighted in red – page number 1 and 2 (lines 16-21 and 75-79).
Comments 2: “Some materials used are unclear, e.g.: “The use of the Likert scale from the QOLIE-31 questionnaire about the patient’s subjective perception of their quality of life” is mentioned in the abstract however in the section “Materials and methods” is unclear how this Likert scale was administered to the patients and the variables referred to”
Response 2: Thank you, a lot, for this comment.
Like the above explained, the “Likert scale” from the QOLIE 31 was used as a measure of subjective ratings of the patients QoL and they choose the answer in the questionnaire based on their subjective feeling about their life quality. The extension of our research plans to analyze the difference between the subjective measure and the overall score of the quality of life, measured using the overall questionnaire score.
We have improved the “Limitations”, “Abstract” and the “Material and methods” part of the manuscript with more clear information, which are highlighted in red – page number 6, 1 and 2 (lines 206-209, lines 16-21 and lines 75-79, respectively).
Comments 3: “Some statistical analysis procedures are unclear, e.g.
We read that the association of various clinical and lifestyle factors with the subjective QoL score were researched. What QoL subjective score was used for the regression analysis and why a score greater>7 was considered as a satisfactory QoL score?”
Response 3: We used the value of 7 points as a cut off as suggested by previous relevant data (Walther K, Volbers B, Erdmann L, Kurzbuch K, Lang JD, Mueller TM, Reindl C, Schwarz M, Schwab S, Hamer HM. Psychosocial long-term outcome in patients with psychogenic non-epileptic seizures. Seizure. 2020 Dec;83:187-192.), where a good QOL was represented as ≥7 score.
We have stated this information and reference in the “Material and methods” section of the manuscript, which are highlighted in red – page number 2 (lines 75-79).
Comments 4: “Finally, the PNES term has currently been replaced in the literature with the “Functional seizures (FS)” term, we suggest using this last proposed terminology in your paper.”
Response 4: Thank you for addressing this issue, we have corrected the term in the whole manuscript.
Reviewer 3 Report
Comments and Suggestions for Authors
The authors studied the overall quality of life of patients with psychogenic non-epileptic seizures (PNES) and evaluated the factors associated with quality-adjusted life years (QALY). The introduction clearly outlines the aims and significance of the study. However, the methods section requires improvement:
- Should the word “conducted” be replaced with “collected” in the following sentence?
"The data was conducted from the local hospital’s information systems ‘Infomedis’ and ‘Heliant’." - The description of sample exclusion is unclear, and the numbers don’t add up to 178. Please include a flow chart for the sample selection process.
- Multinomial logistic regression is typically used when the outcome variable has more than two categories. However, the outcome in this study appears to have only two categories, with 7 as the threshold. Could you clarify why this method was chosen?
- The clinical and lifestyle factors evaluated by this study are not clearly presented. I recommend listing all the factors evaluated in a supplemental table. Additionally, please include a supplementary table showing the results of all regression analyses. It is acceptable to present only the significant factors in the main text if there are too many to include.
The results section could also be improved. In addition to reporting statistical significance, it is important to describe the direction of the associations.
Lastly, while the discussion section is consistent with the results and compares findings with previous studies, it would benefit from a mention of the study's limitations.
Author Response
Comments 1: “The authors studied the overall quality of life of patients with psychogenic non-epileptic seizures (PNES) and evaluated the factors associated with quality-adjusted life years (QALY). The introduction clearly outlines the aims and significance of the study. However, the methods section requires improvement:
- Should the word “conducted” be replaced with “collected” in the following sentence?
"The data was conducted from the local hospital’s information systems ‘Infomedis’ and ‘Heliant’.”
Response 1: Thank you for addressing this issue, we have corrected the term in the text, page 2, line 63-65.
Comments 2: “2. The description of sample exclusion is unclear, and the numbers don’t add up to 178. Please include a flow chart for the sample selection process.”
Response 2: Thank you for this useful suggestion, we have corrected the “Material and methods” section of the manuscript and added Figure 1 (flow chart) in the text, which can be seen at page 2 (lines 66-72) and page 8 (flow chart).
Comments 3: “3.Multinomial logistic regression is typically used when the outcome variable has more than two categories. However, the outcome in this study appears to have only two categories, with 7 as the threshold. Could you clarify why this method was chosen?”
Response 3: Thank you for this question.
We have corrected the text in the previous manuscript version, so the correct sentence is: “Multivariable logistic regression with the selected variables was used to predict outcomes where QoL was greater than 7. All hypothesis tests were two-tailed, with a significance threshold set at a p-value of 0.05.”, which can be seen on page 2, lines 95-97.
Comments 4: “4. The clinical and lifestyle factors evaluated by this study are not clearly presented. I recommend listing all the factors evaluated in a supplemental table. Additionally, please include a supplementary table showing the results of all regression analyses. It is acceptable to present only the significant factors in the main text if there are too many to include.”
Response 4: Thank you for this kind suggestion. We have entered the supplemental Table as requested.
Comments 5: “The results section could also be improved. In addition to reporting statistical significance, it is important to describe the direction of the associations.”
Thank you for this comment.
A multivariable logistic regression model was then refitted using the selected variables, revealing that six (age at FS onset, absence of psychiatric testing, perceiving oneself as mentally changed due to the disease, seizure severity, depressive symptoms, and fear of therapy side effects) were statistically significantly associated with the outcome. All six variables were negatively associated with the outcome, as indicated by odds ratios less than 1.
This is now more clearly stated in the “Results” section on page 4, from lines 121-127.
Comments 6: “Lastly, while the discussion section is consistent with the results and compares findings with previous studies, it would benefit from a mention of the study's limitations.”
Thank you. We have corrected the “Discussion” part and added the “Limitations” section at page 6, lines 206-209.